# How Weakly Supervised Information helps Contrastive Learning

## Abstract

Contrastive learning has shown outstanding performances in both supervised and unsupervised learning. However, little is known about when and how weakly supervised information helps improve contrastive learning, especially from the theoretical perspective. The major challenge is that the existing theory of contrastive learning based on supervised learning frameworks failed to distinguish between supervised and unsupervised contrastive learning. Therefore, we turn to the unsupervised learning frameworks, and based on the posterior probability of labels, we translate the weakly supervised information into a similarity graph under the framework of spectral clustering. In this paper, we investigate two typical weakly supervised learning problems, noisy label learning, and semi-supervised learning, and analyze their influence on contrastive learning within a unified framework. Specifically, we analyze the effect of weakly supervised information on the augmentation graph of unsupervised contrastive learning, and consequently on its corresponding error bound. Numerical experiments are carried out to verify the theoretical findings.

## 1 Introduction

Contrastive learning has shown state-of-the-art empirical performances in both supervised and unsupervised learning. In unsupervised learning, contrastive learning algorithms (Chen et al., 2020; He et al., 2020; Chen et al., 2021; Chen and He, 2021) learn good representations of high-dimensional observations from a large amount of unlabeled data, by pulling together an anchor and its augmented views in the embedding space. On the other hand, supervised contrastive learning (Khosla et al., 2020) uses same-class examples and their corresponding augmentations as positive labels, and achieves significantly better performance than the state-of-the-art cross entropy loss, especially on large-scale datasets.

Recently, contrastive learning has been introduced to solve weakly supervised learning problems such as noisy label learning (Tan et al., 2021; Wang et al., 2022) and semi-supervised learning. For noisy label learning, most methodological studies use contrastive learning as a tool to select confident samples based on the learned representations (Yao et al., 2021; Ortego et al., 2021; Li et al., 2022), whereas the theoretical studies focus on proving the robustness of downstream classifiers with features learned by self-supervised contrastive learning (Cheng et al., 2021; Xue et al., 2022). For semi-supervised learning, contrastive loss is often used as a regularization to improve the precision of pseudo labeling (Lee et al., 2022; Yang et al., 2022).

However, none of the existing studies use weakly supervised information to improve contrastive learning. Perhaps the closest attempt is Yan et al. (2022), which leverages the negative correlations from the noisy data to avoid same-class negatives for contrastive learning. Nonetheless, there are purely empirical results presented, without showing when and how the weakly supervised information helps improve contrastive learning. Moreover, a proper theoretical framework of weakly supervised contrastive learning is especially lacking.

The major challenge lies in the fact that the existing theoretical frameworks compatible with both supervised and unsupervised contrastive learning (Arora et al., 2019; Nozawa and Sato, 2021; Ash et al., 2022; Bao et al., 2022) fail to distinguish between the two settings. To be specific, in order to build a relationship with supervised learning losses, such studies assume that the positive pairs for *unsupervised* contrastive learning are generated from the same latent class, and this is exactly how positive samples for *supervised* contrastive learning are selected. Consequently, such mathemati-

cal modeling cannot tell the difference between supervised and unsupervised contrastive learning. Therefore, in this paper, we in turn base our theoretical analysis on an unsupervised learning framework. Based on the posterior probability of labeled samples, we translate the weakly supervised information into a similarity graph under the framework of spectral clustering. This enables us to analyze the effect of the label information on the augmentation graph of the unsupervised spectral clustering (HaoChen et al., 2021), and consequently on its corresponding error bound.

The contributions of this paper are summarized as follows.

- We for the first time establish a theoretical framework for weakly supervised learning contrastive learning, which is compatible with both noisy label and semi-supervised learning.

- By formulating the label information into a similarity graph based on the posterior probability of labels, we derive the downstream error bound of contrastive learning from both weakly supervised labels and feature information. We show that both noisy labels and semi-supervised labels can improve the error bound of unsupervised contrastive learning under certain constraints on the noise rate and labeled sample size.

- We empirically verify our theoretical results.

## 2 RELATED WORKS

**Theoretical Frameworks of Contrastive Learning.** The theoretical frameworks of unsupervised contrastive learning can be divided into two major categories. The first category is devoted to building the relationship between unsupervised contrastive learning and supervised downstream classification. Arora et al. (2019) first introduces the concept of latent classes, hypothesize that semantically similar points are sampled from the same latent class, and proves that the unsupervised contrastive loss serves an upper bound of downstream supervised learning loss. Nozawa and Sato (2021); Ash et al. (2022); Bao et al. (2022) further investigate the effect of negative samples, and establish surrogate bounds for the downstream classification loss that better match the empirical observations on the negative sample size. However, studies in this category have to assume the existence of supervised latent classes, and that the positive pairs are conditionally independently drawn from the same latent class. This assumption fails to distinguish between supervised and unsupervised contrastive learning, and cannot be used to analyze the weakly supervised setting.

Another major approach is to analyze contrastive learning by modeling the feature similarity. HaoChen et al. (2021) first introduces the concept of the *augmentation graph* to represent the feature similarity of the augmented samples, and analyzes contrastive learning from the perspective of spectral clustering. Shen et al. (2022) uses a stochastic block model to analyze spectral contrastive learning for the problem of unsupervised domain adaption. Similarly, Wang et al. (2021) proposes the concept of *augmentation overlap* to formulate how the positive samples are aligned. Moreover, contrastive learning is also understood through other existing theoretical frameworks of unsupervised learning, such as nonlinear independent component analysis (ICA) (Zimmermann et al., 2021), neighborhood component analysis (NCA) (Ko et al., 2022), variational autoencoder (VAE) (Aitchison, 2021), etc.

In this paper, we follow the second category of contrastive learning approaches, and formulate the weakly supervised information into a similarity graph based on both label and feature information.

**Contrastive Learning for Noisy Label Learning.** Ghosh and Lan (2021) first finds that pretraining with contrastive learning improves robustness to label noise through empirical evidences. Many methodological studies are carried out for noisy label learning with the help of contrastive learning. Yao et al. (2021); Ortego et al. (2021); Li et al. (2022) use representations learned from unsupervised contrastive learning to filter out confident samples from all noisy ones, and in turn use the confident samples to conduct supervised contrastive learning to generate better representations. By contrast, Yan et al. (2022) follows the idea of negative learning (Kim et al., 2019; 2021), and leverage the negative correlations from the noisy data to avoid same-class negatives in contrastive learning. For theoretical studies, Cheng et al. (2021) analyzes the robustness of cross-entropy with SSL features, and Xue et al. (2022) proves the robustness of downstream classifier in contrastive learning.

**Contrastive Learning for Semi-supervised Learning.** Lee et al. (2022); Yang et al. (2022) use contrastive regularization to enhance the reliability of pseudo-labeling in semi-supervised learning.

Kim et al. (2021) introduces a semi-supervised learning method that combines self-supervised contrastive pre-training and semi-supervised fine-tuning based on augmentation consistency regularization. Zhang et al. (2022) uses contrastive loss to model pairwise similarities among samples, generates pseudo labels from the cross entropy loss, and in turn calibrates the prediction distribution of the two branches.

For both noisy label learning and semi-supervised learning tasks, the existing studies all focus on using contrastive learning as a tool to improve the weakly supervised learning performance, whereas to the best of our knowledge, none of the previous works show the effect of weak supervision to contrastive learning itself. To fill in the blank, in this paper, we establish a theoretical framework for weakly supervised contrastive learning, which is compatible with both noisy label and semi-supervised learning tasks.

## 3 PRELIMINARIES

**Notations.** Suppose that random variables $\bar{X} \in \bar{\mathcal{X}} := \mathbb{R}^d$, and $Y \in [r] := \{1, \ldots, r\}$. Let the input natural data $\{(\bar{x}_i, y_i)\}_{i \in [N]}$ be i.i.d. sampled from the joint distribution $\mathrm{P}(\bar{X}, Y)$. Given a natural data $\bar{x} \in \bar{\mathcal{X}}$, we use $\mathcal{A}(\cdot|\bar{x})$ to denote the distribution of its augmentations and use $\mathcal{X}$ to denote the set of all augmented data, which is assumed to be finite but exponentially large. Denote $n = |\mathcal{X}|$.

### 3.1 SPECTRAL CONTRASTIVE LEARNING

In HaoChen et al. (2021), an augmentation graph $\mathcal{G}$ is used to describe the distribution of augmented samples, where the edge weight $w_{xx'} := \mathbb{E}_{\bar{x} \sim \bar{\mathcal{P}}}[\mathcal{A}(x|\bar{x})\mathcal{A}(x'|\bar{x})]$ denotes the marginal probability of generating augmented views $x$ and $x'$ from the same natural data. Due to the total probability mass, $\sum_{x,x' \in \mathcal{X}} w_{xx'} = 1$. The adjacent matrix of the augmentation graph is denoted as $\boldsymbol{A} := (w_{xx'})_{x,x' \in \mathcal{X}} \in \mathbb{R}^{n \times n}$, and the normalized adjacent matrix is denoted as $\bar{\boldsymbol{A}} := D^{-1/2}\boldsymbol{A}D^{-1/2}$, where $D := \mathrm{diag}(w_x)_{x \in \mathcal{X}}$, and $w_x := \sum_{x' \in \mathcal{X}} w_{xx'}$.

In this paper, we consider the spectral contrastive loss proposed by HaoChen et al. (2021), that is, for an embedding function $f : \mathcal{X} \to \mathbb{R}^k$,

$$\mathcal{L}(f) := -2 \cdot \mathbb{E}_{x,x^+}[f(x)^\top f(x^+)] + \mathbb{E}_{x,x'}\left[\left(f(x)^\top f(x')\right)^2\right]. \tag{1}$$

Spectral contrastive loss is proved to be equivalent to the matrix factorization loss, i.e. for $F \in \mathbb{R}^{n \times k} := (u_x)_{x \in \mathcal{X}}, u_x := w_x^{1/2} f(x)$,

$$\mathcal{L}_{\mathrm{mf}}(F) := \|\bar{\boldsymbol{A}} - FF^\top\|_F^2 = \mathcal{L}(f) + \mathrm{const}. \tag{2}$$

### 3.2 NOISY LABEL LEARNING

Recall that we denote the true label of an given instance $x \in \mathcal{X}$ is $y$. One common assumption of the generation procedure of label noise is as follows. Given the true labels, the noisy label is randomly flipped to another label $\tilde{y}$ with some probability. In this paper, we take the widely adopted symmetric label noise assumption as an example.

For notational simplicity, we write the the symmetric label noise assumption in matrix form. Denote $\boldsymbol{Y} := (\eta_j(x_i))_{i \in [n], j \in [r]}, \eta_j(x) = \mathrm{P}(Y = j|x)$, as the posterior probability matrix of the clean label distribution, and denote $\tilde{\boldsymbol{Y}} := (\tilde{\eta}_j(x_i))_{i \in [n], j \in [r]}, \tilde{\eta}_j(x) = \mathrm{P}(\tilde{Y} = j|x)$, as the noisy label distribution. In Assumption 1, we assume that the flipping probability is conditional independent of the input data, and that the flipping probability to all other classes are uniformly at random.

**Assumption 1.** *For symmetric label noise with noise rate $\gamma \in (0, 1)$, we denote the transition matrix* $\boldsymbol{T} = (t_{i,j})_{i \in [r], j \in [r]}$, *where*

$$t_{i,i} = 1 - \gamma, \text{ and } t_{i,j} = \frac{\gamma}{r-1}, \text{ for } j \neq i. \tag{3}$$

*Then the noisy label posterior distribution is assumed to be*

$$\tilde{\boldsymbol{Y}} = \boldsymbol{Y}\boldsymbol{T}. \tag{4}$$

Under Assumption 1, $\boldsymbol{T}$ is symmetric. Specifically, when $\gamma = 0$, $\boldsymbol{T}$ degenerates to the identity matrix $\boldsymbol{I}_{r \times r}$. Moreover, to guarantee the PAC-learnability, we usually assume the true label is the dominating class, i.e. $\gamma < \frac{r-1}{r}$.

### 3.3 SEMI-SUPERVISED LEARNING

For $j \in [r]$, let $n_j$ be the number of labeled samples of Class $j$. Let $n_L = \sum_{j \in [r]} n_{L,j}$ be the number of all labeled samples, and $n_U$ be the number of unlabeled samples. Obviously, we have $n_L + n_U = n$. Usually, the number of labeled samples is much smaller than that of the unlabeled because human annotation is costly and labor-intensive. That is, we can naturally assume $n_L \ll n_U$.

In the following parts of the paper, we analyze the settings of noisy label learning and semi-supervised learning in a unified framework. Without loss of generality, we assume $(x_1, \ldots, x_{n_L})$ is labeled with noise rate $\gamma \in [0, \frac{r-1}{r})$, and denote the corresponding clean and noisy posterior probability matrices as $Y_L$ and $\tilde{Y}_L$, respectively. Then we have $\tilde{Y}_L = Y_L T$. Specifically, when $\gamma = 0$, our analyzing framework degenerates to the standard setting of semi-supervised learning, and when $n_L = n$, our analyzing framework reduces to the standard noisy label learning.

## 4 MATHEMATICAL FORMULATIONS

We mention that our formulation of "similarity graph" is not a distributional assumption on the underlying similarity among data, but to formulate a possible probability of drawing positive samples in contrastive learning that takes both label and feature information into consideration. Specifically, in Sections 4.1 and 4.2, we only discuss the similarity graph describing the weakly supervised labels and neglected feature similarity. Then in Section 4.3, we take both label and feature similarity into consideration through convex combination.

### 4.1 SIMILARITY GRAPH DESCRIBING NOISY LABEL INFORMATION

To leverage the labeled information in the form of similarity graph, we first consider a simple example where noise rate $\gamma = 0$ and the label distribution is deterministic, i.e. for a sample $x$ with true label $y$, the posterior probability $\eta_y(x) = 1$ and $\eta_j(x) = 0$ for $j \neq y$. In this case, we can naturally assume that in the similarity graph describing label information, the intra-class vertices are fully connected and the inter-class vertices are disconnected. That is, $w_{xx'} = 1$ if $x$ and $x'$ has the same label and otherwise $w_{xx'} = 0$.

Then we consider the more general stochastic label scenario. Recall that for unsupervised spectral contrastive learning, the edge weight $w_{xx'}$ in an augmentation graph $\mathcal{G}$ describes the marginal probability of generating $x$ and $x'$ from the same natural data. That is, $w_{xx'}$ describes the joint probability of a pair of positive samples. Similarly, since the positive samples for supervised contrastive learning (Khosla et al., 2020) are selected as all same-class samples, we can naturally define the edge weight $w_{xx'}$ as the probability of two views $x$ and $x'$ generating from the same class, i.e. $w_{xx'} = \sum_{j \in [K]} \eta_j(x)\eta_j(x')$, and therefore $A_L := Y_L Y_L^\top$. Moreover, we denote $\bar{A}$ as the normalized adjacent matrix. For the simplicity of notations, we consider the case where the data is class-balanced, i.e. $n_1 = \ldots = n_r = n_L/r$. Then we have $\bar{A} = \frac{r}{n_L} A$.

Next, we add label noise to the our mathematical formulations. To be specific, when performing supervised contrastive learning based on noisy labeled data, we naturally select positive samples as the samples with the same *noisy* labeled data. According to Assumption 1, we have $\tilde{Y}_L = Y_L T$, where $T$ is symmetric. Then the adjacent matrix of the similarity graph induced by noisy labels is formulated as

$$\tilde{A}_L := \tilde{Y}_L \tilde{Y}_L^\top = Y_L T (Y_L T)^\top = Y_L T T^\top Y_L^\top = Y_L T^2 Y_L^\top. \tag{5}$$

Similarly, when data is class balanced, we have the normalized adjacent matrix $\bar{\tilde{A}}_L = \frac{n_L}{r} \tilde{A}_L$.

### 4.2 SIMILARITY GRAPH DESCRIBING SEMI-SUPERVISED NOISY LABEL INFORMATION

Under the setting of semi-supervised learning, we have no prior knowledge about the label information of the unlabeled samples. From the perspective of unsupervised contrastive learning, the unlabeled samples can be viewed as having unique class labels. Therefore, to construct the similarity graph, we attach sample-specific labels to the unlabeled samples. Thus, the posterior probability matrix of unlabeled samples $Y_U$ is an identity matrix $I_{n_U \times n_U}$. Note that here we only discuss the similarity graph describing supervised information, so the feature similarity between samples is not included in the similarity graph.

Combining both labeled and unlabeled samples, the posterior probability matrix of all semi-supervised samples can be denoted as

$$
\tilde{\boldsymbol{Y}} = \begin{bmatrix} \tilde{\boldsymbol{Y}}_L & \boldsymbol{0} \\ \boldsymbol{0} & \tilde{\boldsymbol{Y}}_U \end{bmatrix} = \begin{bmatrix} \boldsymbol{Y}_L \boldsymbol{T} & \boldsymbol{0} \\ \boldsymbol{0} & \boldsymbol{I}_{n_U \times n_U} \end{bmatrix}. \tag{6}
$$

Therefore, the similarity graph of samples with $n_L$ noisy labels can be denoted as

$$
\tilde{\boldsymbol{A}} = \tilde{\boldsymbol{Y}} \tilde{\boldsymbol{Y}}^\top = \begin{bmatrix} \boldsymbol{Y}_L \boldsymbol{T}^2 \boldsymbol{Y}_L^\top & \boldsymbol{0} \\ \boldsymbol{0} & \boldsymbol{I}_{n_U \times n_U}. \end{bmatrix} \tag{7}
$$

In Lemma 1 we present the influence of symmetric label noise with noise rate $\gamma$ on the similarity graph $\tilde{\boldsymbol{A}}$.

**Lemma 1.** *Under Assumption 1, if the data is class balanced, i.e. $n_1 = \ldots = n_r = \frac{n_L}{r}$, then there holds*

$$
\bar{\tilde{\boldsymbol{A}}} = \begin{bmatrix} \alpha(\gamma) \bar{\boldsymbol{A}}_L + \beta(\gamma) \frac{r}{n_L} \vec{1}_{n_L} \vec{1}_{n_L}^\top & \boldsymbol{0} \\ \boldsymbol{0} & \boldsymbol{I}_{n_U \times n_U} \end{bmatrix}, \tag{8}
$$

*where $\alpha(\gamma) := \left(1 - \frac{r}{r-1}\gamma\right)^2$ and $\beta(\gamma) := \frac{\gamma}{r-1}\left(2 - \frac{r}{r-1}\gamma\right)$.*

Note that without label noise, i.e. $\gamma = 0$, we have $\alpha(\gamma) = 1$ and $\beta(\gamma) = 0$. For the sake of simplicity, in the following we write $\alpha$ and $\beta$ instead of $\alpha(\gamma)$ and $\beta(\gamma)$ when no ambiguity aroused.

In Lemma 1, we show that the effect of symmetric label noise is to add a uniform weight to the edges between all labeled samples. This uniform weight increase the confusion between intra- and inter-class similarities. For example, under the deterministic label scenario, we have $\boldsymbol{A}_L = \boldsymbol{I}_{n_L \times n_L}$. The original intra-class similarity is uniformly shrinked from 1 to $\alpha$ and the inter-class similarity increases from 0 to $\beta$. Moreover, as the noise rate $\gamma$ increases, $\alpha$ decreases and $\beta$ increases, which results in severer confusion between the intra- and inter-class similarities.

### 4.3 SIMILARITY GRAPH DESCRIBING BOTH LABEL AND FEATURE INFORMATION

In this part, we take both label and feature information into consideration. For the feature information, we denote $\boldsymbol{A}_0$ as the augmentation graph of arbitrary unlabeled samples. For label information, we take the similarity graph describing the semi-supervised noisy labeled (augmented) samples $\tilde{\boldsymbol{A}}$ as denoted in equation 7. When leveraging the use of both label and feature information in contrastive learning, we mix the two similarity graphs by convex combination, i.e. for $\theta \in (0, 1)$, the mixed similarity graph of all augmented samples $\boldsymbol{A}_{\theta, \gamma, n_L}$ is denoted as

$$
\boldsymbol{A}_{\theta, \gamma, n_L} := (1 - \theta)\bar{\boldsymbol{A}}_0 + \theta \bar{\tilde{\boldsymbol{A}}}, \tag{9}
$$

where $\bar{\boldsymbol{A}}_0$ and $\bar{\tilde{\boldsymbol{A}}}$ denote the (by row) normalization of $\boldsymbol{A}_0$ and $\tilde{\boldsymbol{A}}$, respectively. Recall that edge weights of the similarity graphs represent the probability of two augmented views being drawn as a pair of positive samples. The similarity graph in equation 9 can be understood as selecting positive pairs based on both weakly supervised labels and feature information.

## 5 THEORETICAL RESULTS

In this section, we first compute eigenvalues of the similarity graph describing both label and feature information, which plays a key role in deriving the error bound of contrastive learning. Then in Section 5.3, we show that the weakly supervised information helps reduce the error of the best possible linear classifier on the representations learned by weakly supervised contrastive learning.

### 5.1 EIGENVALUES OF SIMILARITY GRAPH DESCRIBING WEAKLY SUPERVISED LABEL INFORMATION

We first compute the eigenvalues of the similarity graph describing the semi-supervised noisy labels.

**Proposition 1.** *For arbitrary* $\boldsymbol{Y}$*, assume that the labeled data is class-balanced, i.e.* $\sum_{i \in [n_L]} \eta_j(x_i) = n_L/r$ *for* $j \in [r]$*. Assume that the eigenvalues of* $\bar{\boldsymbol{A}}_L$ *are* $\mu_1, \ldots, \mu_n$ *(in descending order). Then under Assumption 1, the eigenvalues of* $\bar{\bar{\boldsymbol{A}}}$ *are*

$$\tilde{\mu}_1 = \ldots = \tilde{\mu}_{n_U+1} = 1, \tag{10}$$

$$\tilde{\mu}_j = \mu_j \alpha = \mu_j \left(1 - \frac{r}{r-1}\gamma\right)^2, \text{ for } j = n_U + 2, \ldots, n. \tag{11}$$

In Proposition 1, we show that the eigenvalues of $\bar{\bar{\boldsymbol{A}}}$ rely on the eigenvalues of $\bar{\boldsymbol{A}}$ and consequently rely on the posterior probabilities of clean labels. Specifically, if the true label has higher posterior probability, i.e. $\max_{j \in [r]} \mathrm{P}(Y = j|x)$ is larger, then the eigenvalues of $\bar{\boldsymbol{A}}_L$ are larger. On the other hand, the existence of label noise uniformly shrink the eigenvalues of $\bar{\bar{\boldsymbol{A}}}$ except for the largest ones, and larger noise rate $\gamma$ results in smaller $\alpha$ and thus leads to smaller eigenvalues of $\bar{\bar{\boldsymbol{A}}}$. Moreover, the number of largest eigenvalues is decided by the number of unlabeled samples.

Note that $\mathrm{rank}(\tilde{\boldsymbol{A}}) \leq \mathrm{rank}(\boldsymbol{Y}_L) + n_U \leq n_U + r$, and therefore we have $\tilde{\mu}_{n_U+r+1} = \ldots = \tilde{\mu}_n = 0$. Specifically, under the deterministic label scenario, we have $\mu_{n_U+2} = \ldots = \mu_{n_U+r} = 1$. Then the eigenvalues of $\bar{\bar{\boldsymbol{A}}}$ become

$$\tilde{\mu}_1 = \ldots = \tilde{\mu}_{n_U+1} = 1, \tag{12}$$

$$\tilde{\mu}_{n_U+2} = \ldots = \tilde{\mu}_{n_U+r} = \alpha = \left(1 - \frac{r}{r-1}\gamma\right)^2, \tag{13}$$

$$\tilde{\mu}_{n_U+r+1} = \ldots = \tilde{\mu}_n = 0. \tag{14}$$

## 5.2 Eigenvalues of Similarity Graph describing both Label and Feature Information

In the following proposition, we discuss the eigenvalues of the mixed similarity graph $\boldsymbol{A}_{\theta,\gamma,n_L}$ describing both weak labels and feature information.

**Proposition 2.** *Denote* $\lambda_1, \ldots, \lambda_n$ *as the eigenvalues of* $\boldsymbol{A}_{\theta,\gamma,n_L}$*. Then given the eigenvalues of* $\bar{\boldsymbol{A}}_0$*, i.e.* $\nu_1, \ldots, \nu_n$ *and the eigenvalues of* $\bar{\boldsymbol{A}}_L$*, i.e.* $\mu_1, \ldots, \mu_{n_L}$*(in descending order), when* $k \leq n_U$*, there holds*

$$\lambda_{k+1} \geq \max\left\{\theta + (1-\theta)\nu_{n_L+k}, \max_{i=n_L+k-r+1,\ldots,n_L+k-1}\{\theta\alpha\mu_{n+k+1-i} + (1-\theta)\nu_i\}, (1-\theta)\nu_{k+1}\right\}, \tag{15}$$

*when* $n_U < k < n_U + r$*, there holds*

$$\lambda_{k+1} \geq \max\left\{\max_{i=n_L+k-r+1,\ldots,n_L+k-1}\{\theta\alpha\mu_{n+k+1-i} + (1-\theta)\nu_i\}, (1-\theta)\nu_{k+1}\right\}, \tag{16}$$

*and when* $k \geq n_U + r$*, there holds*

$$\lambda_{k+1} \geq (1-\theta)\nu_{k+1}. \tag{17}$$

According to Proposition 2, the lower bound of Specifically, under the deterministic scenario, we have for $k \leq n_U$,

$$\lambda_{k+1} \geq \max\left\{\theta + (1-\theta)\nu_{n_L+k}, \theta\alpha + (1-\theta)\nu_{n_L+k+1-r}, (1-\theta)\nu_{k+1}\right\}, \tag{18}$$

for $n_U < k < n_U + r$,

$$\lambda_{k+1} \geq \max\left\{\theta\alpha + (1-\theta)\nu_{n_L+k+1-r}, (1-\theta)\nu_{k+1}\right\}, \tag{19}$$

and for $k \geq n_U + r$,

$$\lambda_{k+1} \geq (1-\theta)\nu_{k+1}. \tag{20}$$

We see that under the deterministic scenario, the lower bound of the $k + 1$-th largest eigenvalue $\lambda_{k+1}$ of $\boldsymbol{A}_{\theta,\gamma,n_L}$ depends on the eigenvalues $\nu_{n_L+k}, \nu_{n_L+k+1-r}$, and $\nu_{k+1}$ of the unsupervised augmentation graph $\boldsymbol{A}_0$. The value of $\lambda_{k+1}$ is also affected by the weighting parameter $\theta$ and the noise rate $\gamma$. A perhaps anti-intuitive conclusion is that when $k$ is larger than $n_U + r$, the lower bound of $\lambda_{k+1}$ is unaffected by the noise rate. That is, when $k$ is large enough, the weak labels will not affect the $k + 1$-largest eigenvalue of the mixed similarity graph.

## 5.3 WEAK SUPERVISION HELPS REDUCE ERROR BOUND

Recall that the goal of contrastive representation learning is to learn a embedding function $f : \mathcal{X} \to \mathbb{R}^k$. The quality of the learned embedding is often evaluated through linear evaluation. To be specific, denote $B \in \mathbb{R}^{k \times r}$ as the weights of the downstream linear classifier, and the linear predictor is denoted as $\bar{g}_{f,B}(\bar{x}) = \arg\max_{i \in [r]} P_{x \sim \mathcal{A}(\cdot|\bar{x})}(g_{f,B}(x) = i)$, where $g_{f,B}(x) = \arg\max_{i \in [r]}(f(x)^\top B)_i$. In this paper, we focus on analyzing the error bound of the best possible downstream linear classifier $g_{f_{\text{pop}}^*, B^*}$, where $f_{\text{pop}}^* \in \arg\min_{f:\mathcal{X} \to \mathbb{R}^k} \mathcal{L}(f)$ is the minimizer of the population spectral contrastive loss $\mathcal{L}(f)$ as defined in equation 1, and $B^*$ is the optimal weight for the downstream linear classifier.

Following HaoChen et al. (2021), we assume that the labels are recoverable from augmentations, i.e. we assume there exists a classifier $g$ that can predict label $y(x)$ given input $x$ with error at most $\delta \in (0, 1)$ with function $\hat{y} : \mathcal{X} \to [r]$.

**Assumption 2.** *Let $\mathcal{P}_{\bar{X}}$ be the probability distribution of original input data $\bar{x}$. Denote $x$ as an augmented sample and $y$ is its label. Assume that for some $\delta > 0$, there holds*

$$\mathbb{E}_{\bar{x} \sim \mathcal{P}_{\bar{X}}, x \sim \mathcal{A}(\cdot|\bar{x})} \mathbf{1}[\hat{y}(x) \neq y] \leq \delta, \tag{21}$$

*and*

$$\mathbb{E}_{x \sim \text{Unif}(\mathcal{X})} \mathbf{1}[\hat{y}(x) \neq y] \leq \delta. \tag{22}$$

Compared with Assumption 3.5 in HaoChen et al. (2021), Assumption 2 additionally assume the recoverable of labels taking expectation under the uniform probability distribution. We mention that Assumption 2 is a minor revision of the original assumption. The additional assumption equation 22 does not change the nature of the original idea of label recovery, and will be used to bound the error term of learning from weakly supervised labels.

Then in the following theorem we can derive the error bound of downstream linear evaluation learned by weakly supervised contrastive learning.

**Theorem 1.** *For arbitrary $Y$, assume that the labeled data is class-balanced, i.e. $\sum_{i \in [n_L]} \eta_j(x_i) = n_L/r$ for $j \in [r]$. Denote $\nu_1, \ldots, \nu_n$ as the eigenvalues of $\bar{A}_0$ (in descending order). Denote $\mathcal{E} := P_{\bar{x} \sim \mathcal{P}_{\bar{X}}, x \sim \mathcal{A}(\cdot|\bar{x})}(\bar{g}_{f_{\text{pop}}^*, B^*}(x) \neq y(\bar{x}))$ as the linear evaluation error, where $B^* \in \mathbb{R}^{r \times k}$ with norm $\|B^*\|_F \leq 1/\lambda_k$. Then under the deterministic scenario and Assumptions 1 and 2, for $k \leq n_U$, there holds*

$$\mathcal{E} \leq \frac{2[2\delta + \theta(1-\alpha)\frac{r-1}{r}]}{\min\{(1-\theta)(1-\nu_{n_L+k}), (1-\theta)(1-\nu_{n_L+k+1-r}) + \theta(1-\alpha), (1-\theta)(1-\nu_{k+1}) + \theta\}} + 8\delta, \tag{23}$$

*for $n_U + 1 \leq k \leq n_U + r - 1$, there holds*

$$\mathcal{E} \leq \frac{2[2\delta + \theta(1-\alpha)\frac{r-1}{r}]}{\min\{(1-\theta)(1-\nu_{n_L+k+1-r}) + \theta(1-\alpha), (1-\theta)(1-\nu_{k+1}) + \theta\}} + 8\delta, \tag{24}$$

*and for $k \geq n_U + r$, there holds*

$$\mathcal{E} \leq \frac{2[2\delta + \theta(1-\alpha)\frac{r-1}{r}]}{(1-\theta)(1-\nu_{k+1}) + \theta} + 8\delta, \tag{25}$$

*where $\alpha := \left(1 - \frac{r}{r-1}\gamma\right)^2$.*

By Theorem 1, we show that the form of the linear evaluation error bound depends on the dimension of embedding $k$, whereas the error bound is larger when the noise rate $\gamma$ and the label recovery error $\delta$ gets larger, regardless of the dimension $k$. Recall that in HaoChen et al. (2021), the error bound of purely unsupervised contrastive learning is $\frac{4\delta}{1-\nu_{k+1}} + 8\delta$. under the setting of standard semi-supervised classification, i.e. when $\gamma = 0$, and usually $k \leq n_U$, we have

$$\mathcal{E} \leq \frac{4\delta}{1 - \nu_{n_L+k}} + 8\delta, \tag{26}$$

which improves the error bound of purely unsupervised contrastive learning since $\nu_{k+1} \geq \nu_{n_L+k}$.

Next, we discuss the situation when noisy label exists, i.e. $\gamma > 0$.

- For $k \geq n_U + r$,
  - if $1 - \alpha > \frac{2r\delta}{r-1} \frac{\nu_{k+1}}{1-\nu_{k+1}}$, then when $\theta = 0$, there holds

$$\mathcal{E} \leq \frac{2\delta}{1 - \nu_{k+1}} + 8\delta; \tag{27}$$

  - if $1 - \alpha \leq \frac{2r\delta}{r-1} \frac{\nu_{k+1}}{1-\nu_{k+1}}$, then when $\theta = 1$, there holds

$$\mathcal{E} \leq \frac{r-1}{r}(1 - \alpha) + 10\delta. \tag{28}$$

- For $n_U + 1 \leq k \leq n_U + r - 1$, or $k \leq n_U$ and $\delta \leq \frac{r-1}{2r}(1 - \nu_{n_L+k+1-r})$, there holds
  - if $1 - \alpha > \frac{2r\delta}{r-1} \frac{\nu_{k+1}}{1-\nu_{k+1}}$, then when $\theta = 0$,

$$\mathcal{E} \leq \frac{2\delta}{1 - \nu_{k+1}} + 8\delta; \tag{29}$$

  - if $1 - \alpha \leq \frac{2r\delta}{r-1} \frac{\nu_{k+1}}{1-\nu_{k+1}}$, then when $\theta = \frac{\nu_{k+1}-\nu_{n_L+k+1-r}}{\nu_{k+1}-\nu_{n_L+k+1-r}+\alpha}$,

$$\mathcal{E} \leq \frac{2\delta + \frac{r-1}{r}(1 - \alpha)\frac{\nu_{k+1}-\nu_{n_L+k+1-r}}{\nu_{k+1}-\nu_{n_L+k+1-r}+\alpha}}{1 - \frac{\alpha}{\nu_{k+1}-\nu_{n_L+k+1-r}+\alpha}\nu_{k+1}} + 8\delta. \tag{30}$$

We show that when $k < n_U + r$, when $1 - \alpha \leq \frac{2r\delta}{r-1} \frac{\nu_{k+1}}{1-\nu_{k+1}}$, which is equivalent to the noise rate $\gamma$ smaller than a certain threshold, the weakly supervised information can improve the downstream error bound by leveraging both label and feature information and by selecting a proper weighting parameter $\theta$. However, when the noise rate $\gamma$ is large enough, the introduction of noisy labels cannot directly improve the linear evaluation error, regardless of the dimension of the embedding $k$. Fortunately, for contrastive learning under severe label noise, we can use multiple empirical techniques such as using spatial relationship to select confident samples and reduce noise rate, use the pre-filtered weakly supervised data to help improve contrastive learning, and in turn use the improved feature embedding to further reduce label noise. This philosophy has been empirically proved to be effective in many methodology studies (Yao et al., 2021; Ortego et al., 2021; Li et al., 2022).

## 6    EXPERIMENTS

In this section, we aim to empirically verify the our theoretical results that mixing noisy labels and feature information can improve the performance of contrastive learning.

**Loss function.**   Recall that in the theoretical analysis, we investigate the mixed similarity graph $\boldsymbol{A}_{\theta,\gamma,n_L} := (1 - \theta)\bar{\boldsymbol{A}}_0 + \theta\bar{\bar{\boldsymbol{A}}}$. By triangle inequality, we have the matrix factorization loss

$$\mathcal{L}_{\mathrm{mf}}(F) = \|\boldsymbol{A}_{\theta,\gamma,n_L} - FF^\top\|_F^2 = \|(1 - \theta)\bar{\boldsymbol{A}}_0 + \theta\bar{\bar{\boldsymbol{A}}} - FF^\top\|_F^2$$
$$\leq (1 - \theta)\|\bar{\boldsymbol{A}}_0 - FF^\top\|_F^2 + \theta\|\bar{\bar{\boldsymbol{A}}} - FF^\top\|_F^2. \tag{31}$$

According to equation 2, the spectral contrastive loss is equivalent to the matrix factorization loss. Therefore, in experiments, we use the convex combination of supervised and unsupervised contrastive losses to leverage the noisy label and feature information, i.e. for $\theta \in (0, 1)$, we use

$$\mathcal{L}_{\mathrm{mix}} := (1 - \theta)\mathcal{L}_{\mathrm{unsup}} + \theta\mathcal{L}_{\mathrm{sup}}. \tag{32}$$

**Setup.**   We conduct numerical comparisons on the CIFAR-10 and TinyImagenet-200 benchmark image dataset (the results on TinyImagenet-200 can be found in Appendix A.2) and follow the setting of SimCLR (Chen et al., 2020) and SupCon (Khosla et al., 2020). We use the SGD optimizer and use ResNet-50 as the encoder and a 2-layer MLP as the projection head. We run experiments on 4 NVIDIA Tesla v100 32GB GPUs. The data augmentations we use are random crop and resize (with random flip), color distortion and color dropping. The models are trained with batch size 1024

and 1000 epochs for each model. We evaluate the self-supervised learned representation by linear evaluation protocol, where a linear classifier is trained on the top of the encoder, and regard its test accuracy as the performance of the encoder. The symmetric noisy labels are generated by flipping the labels of a given proportion of training samples uniformly to one of the other class labels.

In Table 1, we compare the performance of unsupervised contrastive learning (SimCLR), supervised contrastive learning (SupCon), and weakly supervised contrastive learning (Mix) under noise rate $\gamma = 5\%$ and $\gamma = 20\%$. In SimCLR, we neglect all labels in the training procedure, and in SupCon, we select positive samples as those with the same noisy annotations. The parameter grid of $\theta$ for Mix is $\{0.1, 0.2, 0.4, 0.6, 0.8, 0.9, 0.95\}$. The performance comparisons with more noise rates can be found in Appendix A.2. The best results are marked in **bold**. The standard deviation is also reported.

Table 1: Performance comparisons on CIFAR-10 dataset.

|  | $\gamma = 0\%$ | $\gamma = 5\%$ | $\gamma = 20\%$ |
|---|---|---|---|
| SimCLR | 92.30 | $91.99 \pm 0.06$ | $91.59 \pm 0.06$ |
| SupCon | **95.48** | $93.86 \pm 0.05$ | $90.56 \pm 0.05$ |
| Mix | **95.48** | $\mathbf{94.39 \pm 0.03}$ | $\mathbf{92.84 \pm 0.09}$ |

We show in Table 1 that if the noise rate is small ($\gamma = 5\%$), SupCon results in better performance than SimCLR, whereas when the noise rate raises to $\gamma = 20\%$, the noisy labels actually harm the performance of contrastive learning. Nonetheless, under $\gamma = 5\%$ and $\gamma = 20\%$, the weakly supervised Mix outperforms both unsupervised SimCLR and supervised SupCon. This verifies the result in Theorem 1 that when the noise rate is smaller than a certain threshold, leveraging both weakly supervised and feature information helps improve the performance of contrastive learning.

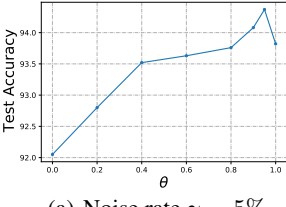 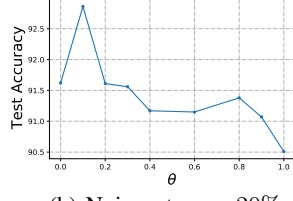

(a) Noise rate $\gamma = 5\%$.      (b) Noise rate $\gamma = 20\%$.

Figure 1: Parameter analysis of $\theta$.

In Figure 1 we conduct parameter analysis of $\theta$ for the weakly supervised contrastive learning (Mix). We show that as $\theta$ increases from 0 to 1, the performance of Mix first increases and then decreases. Moreover, larger noise rate requires smaller optimal $\theta$. The optimal value of $\theta$ for $\gamma = 5\%$ is larger than that for $\gamma = 20\%$. That is, under severer label noise, less supervised information should be considered in weakly supervised contrastive learning.

## 7   CONCLUSION

In this paper, we establish a theoretical framework for weakly supervised contrastive learning, which is compatible with the settings of both noisy label learning and semi-supervised learning. By formulating a mixed similarity graph describing both weakly supervised label information and unsupervised feature information, we analyze the weakly supervised spectral contrastive learning based on the framework of spectral clustering, and derive the downstream linear evaluation error bound. Our theoretical results show that semi-supervised noisy labels improves the downstream error bound when the noise rate is smaller than a certain threshold. Our theoretical framework reveals the effect of weak supervision to contrastive learning, and has the potential to explain the existing weakly supervised learning algorithms based on contrastive learning approaches and to inspire new algorithms. For future works, we will investigate the effect of more complex weak supervision, such as active learning and label-dependent label noise, on contrastive learning.

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

## A APPENDIX

### A.1 PROOFS

*Proof of Lemma 1.* Under Assumption 1, we have

$$
(\boldsymbol{T}^2)_{i,j} = \begin{cases} (1-\gamma)^2 + \gamma^2/(r-1), & i = j \\ 2\gamma(1-\gamma)/(r-1) + (r-2)\gamma^2/(r-1)^2, & i \neq j \end{cases}
$$
$$
= \begin{cases} (1-\gamma)^2 + \gamma^2/(r-1), & i = j \\ \dfrac{\gamma}{r-1}\Big(2 - \dfrac{r}{r-1}\gamma\Big), & i \neq j. \end{cases} \tag{33}
$$

That is, we have

$$
\begin{aligned}
\boldsymbol{T}^2 &= \Big[(1-\gamma)^2 + \gamma^2/(r-1) - \frac{\gamma}{r-1}\Big(2 - \frac{r}{r-1}\gamma\Big)\Big]\boldsymbol{I}_{r\times r} \\
&\quad + \frac{\gamma}{r-1}\Big(2 - \frac{r}{r-1}\gamma\Big)\vec{1}_r\vec{1}_r^\top \\
&= \Big(1 - \frac{r}{r-1}\gamma\Big)^2 \boldsymbol{I}_{r\times r} + \frac{\gamma}{r-1}\Big(2 - \frac{r}{r-1}\gamma\Big)\vec{1}_r\vec{1}_r^\top \\
&:= \alpha\boldsymbol{I}_{r\times r} + \beta\vec{1}_r\vec{1}_r^\top. 
\end{aligned} \tag{34}
$$

Given $\gamma \in [0, 1)$, we have

$$
\begin{aligned}
\tilde{\boldsymbol{A}}_L = \boldsymbol{Y}_L \boldsymbol{T}^2 \boldsymbol{Y}_L^\top &= \boldsymbol{Y}_L\big(\alpha\boldsymbol{I}_{r\times r} + \beta\vec{1}_r\vec{1}_r^\top\big)\boldsymbol{Y}_L^\top \\
&= \alpha\boldsymbol{Y}_L\boldsymbol{Y}_L^\top + \beta\boldsymbol{Y}_L\vec{1}_r\vec{1}_r^\top\boldsymbol{Y}_L^\top \\
&= \alpha\boldsymbol{A}_L + \beta\vec{1}_{n_L}\vec{1}_{n_L}^\top, 
\end{aligned} \tag{35}
$$

where the last equality holds because $\sum_j \eta_j(x_i) = 1$ for $i \in [n]$. $\qquad\square$

and the normalized augmentation graph is

$$
\bar{\tilde{\boldsymbol{A}}} = \tilde{\boldsymbol{D}}^{-1/2}\tilde{\boldsymbol{A}}\tilde{\boldsymbol{D}}^{-1/2}, \tag{36}
$$

where

$$
\tilde{\boldsymbol{D}} = \begin{bmatrix} \tilde{\boldsymbol{D}}_L & \boldsymbol{0} \\ \boldsymbol{0} & \boldsymbol{I}_{n_U \times n_U} \end{bmatrix}, \tag{37}
$$

$$
\tilde{\boldsymbol{D}}_L = \operatorname{diag}(d_i), \tag{38}
$$

and

$$
\begin{aligned}
d_i &= \sum_{j \in [n_L]} \tilde{\boldsymbol{A}}_{i,j} = \alpha \sum_{j \in [n_L]} \sum_{\ell \in [r]} \eta_\ell(x_i)\eta_\ell(x_j) + n_L\beta \\
&= \alpha \sum_{\ell \in [r]} \eta_\ell(x_i) \sum_{j \in [n_L]} \eta_\ell(x_j) + n_L\beta = \alpha \sum_{\ell \in [r]} \eta_\ell(x_i)n_\ell + n_L\beta 
\end{aligned} \tag{39}
$$

Specifically, when the labeled data is class-balanced, i.e. $n_1 = \ldots = n_r = n_L/r$. Then we have

$$
d_i = \frac{n_L}{r}\alpha \sum_{\ell \in [r]} \eta_\ell(x_i) + n_L\beta = \frac{n_L}{r}\alpha + n\beta = \frac{n_L}{r}, \tag{40}
$$

and thus

$$
\bar{\tilde{\boldsymbol{A}}} = \begin{bmatrix} \alpha\frac{r}{n_L}\boldsymbol{A}_L + \beta\frac{r}{n_L}\vec{1}_{n_L}\vec{1}_{n_L}^\top & \boldsymbol{0} \\ \boldsymbol{0} & \boldsymbol{I}_{n_U \times n_U} \end{bmatrix}. \tag{41}
$$

*Proof of Proposition 1.* We first prove that $v_1 = \frac{1}{\sqrt{n_L}}\vec{1}_{n_L}$ is an eigenvector of $\bar{A}_L := \frac{r}{n_L}A_L$ with eigenvalue $\mu_1 = 1$. To be specific,

$$
\begin{aligned}
\bar{A}_L \cdot \frac{1}{\sqrt{n_L}}\vec{1}_{n_L} &= \frac{1}{\sqrt{n_L}} \cdot \frac{r}{n_L} A_L \cdot \vec{1}_{n_L} \\
&= \frac{1}{\sqrt{n_L}} \cdot \frac{r}{n_L} Y_L Y_L^\top \vec{1}_{n_L} \\
&= \frac{1}{\sqrt{n_L}} \cdot \frac{r}{n_L} Y_L \frac{n_L}{r}\vec{1}_r \\
&= \frac{1}{\sqrt{n_L}}\vec{1}_{n_L},
\end{aligned}
\tag{42}
$$

where the second last equality is due to class balance, i.e. $\sum_{i \in [n_L]} \eta_j(x_i) = n_L/r$ for $j \in [r]$, and the last equality holds because $\sum_{j \in [r]} \eta_j(x_i) = 1$ for $i \in [n_L]$.

Therefore, we can rewrite $\bar{A}_L$ as

$$
\bar{A}_L = \left[\frac{1}{\sqrt{n_L}}\vec{1}_{n_L}, v_2, \ldots, v_{n_L}\right]
\begin{bmatrix}
1 & 0 & \cdots & 0 \\
0 & \mu_2 & \cdots & 0 \\
\vdots & \vdots & & \vdots \\
0 & 0 & \cdots & \mu_{n_L}
\end{bmatrix}
\begin{bmatrix}
\frac{1}{\sqrt{n_L}}\vec{1}_{n_L}^\top \\
v_2^\top \\
\vdots \\
v_{n_L}^\top
\end{bmatrix}.
\tag{43}
$$

Note that $\frac{1}{n_L}\vec{1}_{n_L}\vec{1}_{n_L}^\top$ can be decomposed as

$$
\begin{aligned}
\frac{1}{n_L}\vec{1}_{n_L}\vec{1}_{n_L}^\top &= \left(\frac{1}{\sqrt{n_L}}\vec{1}_{n_L}\right)\left(\frac{1}{\sqrt{n_L}}\vec{1}_{n_L}\right)^\top \\
&= \left[\frac{1}{\sqrt{n_L}}\vec{1}_{n_L}, v_2, \ldots, v_{n_L}\right]
\begin{bmatrix}
1 & 0 & \cdots & 0 \\
0 & 0 & \cdots & 0 \\
\vdots & \vdots & & \vdots \\
0 & 0 & \cdots & 0
\end{bmatrix}
\begin{bmatrix}
\frac{1}{\sqrt{n_L}}\vec{1}_{n_L}^\top \\
v_2^\top \\
\vdots \\
v_{n_L}^\top
\end{bmatrix}.
\end{aligned}
\tag{44}
$$

Then we have

$$
\begin{aligned}
\bar{\bar{A}}_L &:= \alpha\frac{r}{n_L}A_L + r\beta\frac{1}{n_L}\vec{1}_{n_L}\vec{1}_{n_L}^\top \\
&= \left[\frac{1}{\sqrt{n_L}}\vec{1}_{n_L}, v_2, \ldots, v_{n_L}\right]
\begin{bmatrix}
\alpha & 0 & \cdots & 0 \\
0 & \alpha\mu_2 & \cdots & 0 \\
\vdots & \vdots & & \vdots \\
0 & 0 & \cdots & \alpha\mu_{n_L}
\end{bmatrix}
\begin{bmatrix}
\frac{1}{\sqrt{n_L}}\vec{1}_{n_L}^\top \\
v_2^\top \\
\vdots \\
v_{n_L}^\top
\end{bmatrix} \\
&\quad \cdot \left[\frac{1}{\sqrt{n_L}}\vec{1}_{n_L}, v_2, \ldots, v_{n_L}\right]
\begin{bmatrix}
r\beta & 0 & \cdots & 0 \\
0 & 0 & \cdots & 0 \\
\vdots & \vdots & & \vdots \\
0 & 0 & \cdots & 0
\end{bmatrix}
\begin{bmatrix}
\frac{1}{\sqrt{n_L}}\vec{1}_{n_L}^\top \\
v_2^\top \\
\vdots \\
v_{n_L}^\top
\end{bmatrix} \\
&= \left[\frac{1}{\sqrt{n_L}}\vec{1}_{n_L}, v_2, \ldots, v_{n_L}\right]
\begin{bmatrix}
\alpha + r\beta & 0 & \cdots & 0 \\
0 & \alpha\mu_2 & \cdots & 0 \\
\vdots & \vdots & & \vdots \\
0 & 0 & \cdots & \alpha\mu_{n_L}
\end{bmatrix}
\begin{bmatrix}
\frac{1}{\sqrt{n_L}}\vec{1}_{n_L}^\top \\
v_2^\top \\
\vdots \\
v_{n_L}^\top
\end{bmatrix}.
\end{aligned}
\tag{45}
$$

Since $\alpha + r\beta = 1$, the eigenvalues of $\bar{\bar{A}}_L$ are $1, \alpha\mu_2, \ldots, \alpha\mu_{n_L}$. Thus the eigenvalues of

$$
\bar{\bar{A}} = \begin{bmatrix}
\bar{\bar{A}}_L & \mathbf{0} \\
\mathbf{0} & I_{n_U \times n_U}
\end{bmatrix}
\tag{46}
$$

are

$$\tilde{\mu}_1 = \ldots = \tilde{\mu}_{n_U+1} = 1, \tag{47}$$

$$\tilde{\mu}_j = \alpha\mu_j, \text{ for } j = n_U + 2, \ldots, n. \tag{48}$$

$\square$

*Proof of Proposition 2.* By equation 13 in Fulton (2000), for two real symmetric $n$ by $n$ matrix $(1-\theta)\bar{A}_0$ and $\theta\tilde{\bar{A}}$, the $k+1$-th largest eigenvalue of $A_{\theta,\lambda,n_L} := (1-\theta)\bar{A}_0 + \theta\tilde{\bar{A}}$ can take any value in the interval

$$\max_{i+j=n+k+1}(1-\theta)\nu_i + \theta\tilde{\mu}_j \leq \lambda_k \leq \min_{i+j=k+2}(1-\theta)\nu_i + \theta\tilde{\mu}_j. \tag{49}$$

By Proposition 1, we have

$$\tilde{\mu}_j = \begin{cases} 1, & j = 1, \ldots, n_U + 1; \\ \alpha\mu_j, & j = n_U + 2, \ldots, n_U + r; \\ 0, & j = n_U + r + 1, \ldots, n. \end{cases} \tag{50}$$

Therefore, we have

$$\max_{i+j=n+k+1}(1-\theta)\nu_i + \theta\tilde{\mu}_j$$

$$= \max_{1\leq i\leq n+k+1}(1-\theta)\nu_i + \theta\tilde{\mu}_{n+k+1-i}$$

$$= \max\begin{cases} \theta + (1-\theta)\nu_i, & i = n_L + k, \ldots, n; \\ \theta\alpha\mu_{n+k+1-i} + (1-\theta)\nu_i, & i = n_L + k - r + 1, \ldots, n_L + k - 1; \\ (1-\theta)\nu_i, & i = k + 1, \ldots, n_L + k - r \end{cases}$$

$$= \max\begin{cases} \theta + (1-\theta)\nu_{n_L+k} \\ \theta\alpha\mu_{n+k+1-i} + (1-\theta)\nu_i, & i = n_L + k - r + 1, \ldots, n_L + k - 1; \\ (1-\theta)\nu_{k+1}, \end{cases} \tag{51}$$

where the last equality holds because $\{\nu_i\}_{i\in[n]}$ is ranked in descending order. Then when $k \leq n_U$,

$$\lambda_{k+1} \geq \max\left\{\theta + (1-\theta)\nu_{n_L+k}, \max_{i=n_L+k-r+1,\ldots,n_L+k-1}\{\theta\alpha\mu_{n+k+1-i} + (1-\theta)\nu_i\}, (1-\theta)\nu_{k+1}\right\}, \tag{52}$$

when $n_U < k < n_U + r$,

$$\lambda_{k+1} \geq \max\left\{(1-\theta)\nu_{k+1}, \max_{i=n_L+k-r+1,\ldots,n_L+k-1}\{\theta\alpha\mu_{n+k+1-i} + (1-\theta)\nu_i\}\right\}, \tag{53}$$

and when $k \geq n_U + r$,

$$\lambda_{k+1} \geq (1-\theta)\nu_{k+1}. \tag{54}$$

$\square$

*Proof of Theorem 1.* By Lemma B.3 of HaoChen et al. (2021), for any labeling function $\vec{y} : \mathcal{X} \to [r]$, there exists a linear probe $B^* \in \mathbb{R}^{r\times k}$ with norm $\|B^*\|_F \leq 1/\lambda_k$ such that

$$\mathrm{P}_{\bar{x}\sim\mathcal{P}_{\bar{X}}, x\sim\mathcal{A}(\cdot|\bar{x})}\left(g_{f^*_{\text{pop}}, B^*}(x) \neq y(\bar{x})\right) \leq \frac{2\phi^{\hat{y}}}{1 - \lambda_{k+1}} + 8\delta, \tag{55}$$

where according to the definition of $A_{\theta,\lambda,n_L}$,

$$\phi^{\hat{y}} = (1-\theta)\sum_{i,j\in[n]}(A_0)_{i,j}\mathbf{1}[\hat{y}(x_i) \neq \hat{y}(x_j)] + \theta\frac{1}{n_L}\sum_{i,j\in[n]}\tilde{\bar{A}}_{i,j}\mathbf{1}[\hat{y}(x_i) \neq \hat{y}(x_j)]. \tag{56}$$

We investigate the RHS of equation 56 respectively. The first term is

$$\sum_{i,j\in[n]}(A_0)_{i,j}\mathbf{1}[\hat{y}(x_i) \neq \hat{y}(x_j)]$$

$$= \sum_{i,j\in[n]} \mathbb{E}_{\bar{x}\sim\mathcal{P}_{\bar{\mathcal{X}}}} \mathcal{A}(x_i|\bar{x})\mathcal{A}(x_j|\bar{x})\mathbf{1}[\hat{y}(x_i)\neq\hat{y}(x_j)]$$

$$\leq \sum_{i,j\in[n]} \mathbb{E}_{\bar{x}\sim\mathcal{P}_{\bar{\mathcal{X}}}} \mathcal{A}(x_i|\bar{x})\mathcal{A}(x_j|\bar{x})\big(\mathbf{1}[\hat{y}(x_i)\neq\hat{y}(\bar{x})] + \mathbf{1}[\hat{y}(x_j)\neq\hat{y}(\bar{x})]\big)$$

$$= 2 \sum_{i,j\in[n]} \mathbb{E}_{\bar{x}\sim\mathcal{P}_{\bar{\mathcal{X}}}} \mathcal{A}(x_i|\bar{x})\mathbf{1}[\hat{y}(x_i)\neq\hat{y}(\bar{x})]$$

$$= 2\delta. \tag{57}$$

The second term is

$$\frac{1}{n_L} \sum_{i,j\in[n]} \bar{\bar{\boldsymbol{A}}}_{i,j}\mathbf{1}[\hat{y}(x_i)\neq\hat{y}(x_j)]$$

$$= \frac{1}{n_L} \sum_{i,j\in[n_L]} (\bar{\bar{\boldsymbol{A}}}_L)_{i,j}\mathbf{1}[\hat{y}(x_i)\neq\hat{y}(x_j)] + \frac{1}{n_L} \sum_{i>n_L} \mathbf{1}[\hat{y}(x_i)\neq\hat{y}(x_i)]$$

$$+ 2\frac{1}{n_L} \sum_{i\leq n_L, j>n_L} \bar{\bar{\boldsymbol{A}}}_{i,j}\mathbf{1}[\hat{y}(x_i)\neq\hat{y}(x_i)]. \tag{58}$$

According to the definition of $\bar{\bar{\boldsymbol{A}}}$, the last two terms are equal to 0. Then by Lemma 1, the second term on the RHS of equation 56 becomes

$$\frac{1}{n_L} \sum_{i,j\in[n]} \bar{\bar{\boldsymbol{A}}}_{i,j}\mathbf{1}[\hat{y}(x_i)\neq\hat{y}(x_j)]$$

$$= \frac{1}{n_L} \sum_{i,j\in[n_L]} \Big(\alpha\bar{\boldsymbol{A}}_{i,j} + \beta\frac{r}{n_L}\vec{\mathbf{1}}_{n_L}\vec{\mathbf{1}}_{n_L}^{\top}\Big)\mathbf{1}[\hat{y}(x_i)\neq\hat{y}(x_j)]$$

$$\leq \alpha\frac{r}{n_L^2} \sum_{i,j\in[n_L]} \sum_{\ell\in[r]} \eta_\ell(x_i)\eta_\ell(x_j)\Big(\mathbf{1}[\hat{y}(x_i)\neq\ell] + \mathbf{1}[\hat{y}(x_j)\neq\ell]\Big)$$

$$+ \beta\frac{r}{n_L^2} \sum_{i,j\in[n_L]} \Big(\mathbf{1}[\hat{y}(x_i)\neq y_i] + \mathbf{1}[\hat{y}(x_j)\neq y_j] + \mathbf{1}[y_i\neq y_j]\Big)$$

$$= \alpha\frac{r}{n_L^2} \sum_{\ell\in[r]} 2\frac{n_L}{r} \sum_{i\in[n_L]} \eta_\ell(x_i)\mathbf{1}[\hat{y}(x_i)\neq\ell]$$

$$+ \beta\frac{r}{n_L^2}\Big(2n_L \sum_{i\in[n_L]} \mathbf{1}[\hat{y}(x_i)\neq y_i] + \sum_{i,j\in[n_L]} \mathbf{1}[y_i\neq y_j]\Big)$$

$$\leq 2\alpha\frac{1}{n_L} \sum_{i\in[n_L]} \sum_{\ell\in[r]} \eta_\ell(x_i)\mathbf{1}[\hat{y}(x_i)\neq\ell] + 2\beta\frac{r}{n_L} \sum_{i\in[n_L]} \mathbf{1}[\vec{y}(x_i)\neq y_i] + \beta(r-1). \tag{59}$$

Under the deterministic scenario, we have

$$\frac{1}{n_L} \sum_{i,j\in[n]} \bar{\bar{\boldsymbol{A}}}_{i,j}\mathbf{1}[\hat{y}(x_i)\neq\hat{y}(x_j)]$$

$$= 2\alpha\frac{1}{n_L} \sum_{i\in[n_L]} \mathbf{1}[\hat{y}(x_i)\neq\ell] + 2\beta r\frac{1}{n_L} \sum_{i\in[n_L]} \mathbf{1}[\vec{y}(x_i)\neq y_i] + \beta(r-1)$$

$$= 2\alpha\delta + 2\beta r\delta + \beta(r-1)$$

$$= 2\alpha\delta + 2(1-\alpha)\delta + (1-\alpha)\frac{r-1}{r}$$

$$= 2\delta + (1-\alpha)\frac{r-1}{r}, \tag{60}$$

where the second last equation holds due to $\alpha + r\beta = 1$. Combining equation 56, equation 57 and equation 60, we have

$$\phi^{\hat{y}} \leq (1-\theta)2\delta + \theta\Big(2\delta + (1-\alpha)\frac{r-1}{r}\Big) = 2\delta + \theta(1-\alpha)\frac{r-1}{r}. \tag{61}$$

Therefore, by equation 55, we have

$$\mathcal{E} := \mathrm{P}_{\bar{x}\sim\mathcal{P}_{\bar{X}},x\sim\mathcal{A}(\cdot|\bar{x})}\Big(g_{f_{\text{pop}}^*,B^*}(x) \neq y(\bar{x})\Big) \leq \frac{2[2\delta + \theta(1-\alpha)\frac{r-1}{r}]}{1-\lambda_{k+1}} + 8\delta, \tag{62}$$

Combined with Proposition 2, we have for $k \leq n_U$, there holds

$$\mathcal{E} \leq \frac{2[2\delta + \theta(1-\alpha)\frac{r-1}{r}]}{\min\{(1-\theta)(1-\nu_{n_L+k}),(1-\theta)(1-\nu_{n_L+k+1-r})+\theta(1-\alpha),(1-\theta)(1-\nu_{k+1})+\theta\}} + 8\delta, \tag{63}$$

for $n_U + 1 \leq k \leq n_U + r - 1$, there holds

$$\mathcal{E} \leq \frac{2[2\delta + \theta(1-\alpha)\frac{r-1}{r}]}{\min\{(1-\theta)(1-\nu_{n_L+k+1-r})+\theta(1-\alpha),(1-\theta)(1-\nu_{k+1})+\theta\}} + 8\delta, \tag{64}$$

and for $k \geq n_U + r$, there holds

$$\mathcal{E} \leq \frac{2[2\delta + \theta(1-\alpha)\frac{r-1}{r}]}{(1-\theta)(1-\nu_{k+1})+\theta} + 8\delta. \tag{65}$$

$\square$

## A.2 ADDITIONAL EXPERIMENTS

We run additional experiments on CIFAR-10 dataset with a noise rate $\gamma$ varying from $0\%$ to $60\%$ in Table 2. The parameter grid of $\theta$ for Mix is $\{0.05, 0.1, 0.2, 0.4, 0.6, 0.8, 0.9, 0.95\}$. Table 2 that our Mix consistently outperforms SimCLR and SupCon.

Table 2: Additional Performance comparisons on CIFAR-10 dataset.

|  | $\gamma = 0\%$ | $\gamma = 5\%$ | $\gamma = 10\%$ | $\gamma = 20\%$ | $\gamma = 30\%$ | $\gamma = 40\%$ | $\gamma = 50\%$ | $\gamma = 60\%$ |
|---|---|---|---|---|---|---|---|---|
| SimCLR | 92.30 | 91.99 | 91.63 | 91.59 | 90.95 | 90.42 | 90.18 | 88.92 |
| SupCon | **95.48** | 93.86 | 92.62 | 90.56 | 87.76 | 84.80 | 80.56 | 74.90 |
| Mix | **95.48** | **94.39** | **93.12** | **92.84** | **91.89** | **91.74** | **90.80** | **90.66** |

We run additional experimental comparisons on the TinyImagenet-200 dataset with noise rate $\gamma = 0.4$. The parameter grid of $\theta$ is $\{0.1, 0.2, 0.4, 0.6\}$. We additionally adopt Gaussian Blur for data augmentation and keep the other experimental setups the same as Table 1. The results are presented in Table 3. We show that Mix outperforms both SimCLR and SupCon on the TinyImagenet-200 dataset.

Table 3: Performance comparisons on TinyImagenet-200 dataset.

|  | $\gamma = 0\%$ | $\gamma = 40\%$ |
|---|---|---|
| SimCLR | $53.78 \pm 0.13$ | $49.79 \pm 0.05$ |
| SupCon | $\mathbf{63.79 \pm 0.23}$ | $50.24 \pm 0.09$ |
| Mix | $\mathbf{63.79 \pm 0.23}$ | $\mathbf{53.40 \pm 0.05}$ |

