# OpenReview forum: "How Weakly Supervised Information helps Contrastive Learning"
_ICLR.cc/2023/Conference — Submitted to ICLR 2023_

### Official Review · Reviewer_v3Am · 2022-10-18

**Confidence:** 3
**Correctness:** 2
**Technical Novelty And Significance:** 2
**Empirical Novelty And Significance:** 3
**Recommendation:** 5

**Clarity, Quality, Novelty And Reproducibility:**

I raised some unclear parts of this paper with minor typos and grammatical suggestions. Addressing them should improve the quality of this paper.

**Important remark**: In Section 4, the authors "formulate" the augmentation/similarity graph for semi-supervised noisy labels together with feature similarity. At a first sight, this formulation looks rather arbitrary. For example, the assumed posterior probability for unlabeled data looks not a natural one without having dependency with labeled data, and the feature similarity $\\mathbf{A}_0$ and label similarity $\\mathbf{A}$ is concatenated by the convex combination. But eventually, I realized that this "formulation" is a kind of the authors' "design" of data augmentation, but not a distributional assumption on the underlying similarity among data.

I misunderstood this point because HaoChen et al. originally introduced the augmentation graph (in the unsupervised setup) and its weights to assume/model the underlying possibility of data augmentation. Thus, I suppose it's important to emphasize that this paper introduces an (apparently arbitrary) matrix $\\mathbf{A}$ to specify how to augment data.

Another (slightly) confusing wording might be the section title of Section 4.1 "Similarity Graph **Induced** By Noisy Labels" (and Sections 4.2 and 4.3 as well). The graph does not seem to be induced from the underlying nature, but the authors specify a convenient one for the latter purpose as far as I understand.

**Minor**
- In the third paragraph of Section 1, "On the other hand" in the fourth line may sound logically unnatural.
- In Section 3.1, `diag` -> `\mathrm{diag}`,  `const` -> `\mathrm{const}`, and $A$ -> $\\mathbf{A}$ (the fifth line).
- In Section 3.3, `<<` -> `\ll` (the fourth line).
- In Section 4.2, "Therefore" in the second line may sound logically unnatural. Such conjunctions appear a little bit too frequently in this paragraph, distracting readers.
- In Section 4.2 (the second last line), "as the noise rate $\\gamma$ increases, $\\alpha$ increases and $\\beta$ decreases" -> "$\\alpha$ **decreases** and $\\beta$ **increases**".
- In Section 4.3 (the second line), "as the augmentation graph arbitrary unlabeled samples" -> "as the augmentation graph **of** arbitrary unlabeled samples." (?)
- In Section 5.1 (three lines after Proposition 1), $\\bar{A}_L$ -> $\\bar{\\mathbf{A}}_L$.
- In Section 5.2 (right after Proposition 2), the text looks strange.
- In Section 5.3, the definition of $g_{f,B}(\bar{x})$ appearing in the fourth line looks a circular definition. The linear projection matrix $\\mathbf{B}$ should be involved in it.
- In Section 5.3 (the fifth line), $\\mathcal{L}(f)$ is needed in the definition of $f_{\\mathrm{pop}}^*$
- In Section 5.3 (Assumption 2), many notations are not defined properly. It is essential to show what $y(x)$, $\hat{y}(x)$, and $\\mathcal{P}_{\\bar{X}}$ are. Moreover, $x_i$ and $y_i$ in equations (21) and (22) are not given yet.
- In Section 5.3 (Theorem 1), why does the definition of the downstream error $\\mathcal{E}$ involve the augmentation $\\mathcal{A}$? I do not think using augmentation in the downstream task is a common practice.
- In Section 5.3 (Theorem 1), "$B^* \\in \\mathbb{R}^{r \\times k}$ with norm $\\|B^*\\|_F \\le 1/\\tilde{\\lambda}_k$" is unclear. It seems that $\\tilde{\\lambda}_k$ has not been defined yet so far. In addition, it is ambiguous whether this norm bound is a newly-introduced assumption or a corollary of the assumptions so far.

**Strength And Weaknesses:**

### Strengths

This paper seems to be the first one to consider how weak supervision helps contrastive learning, not the other way around, how contrastive learning can elicit useful information for weakly supervised learning. In practice, it is a natural scenario to do contrastive learning with a bunch of unreliable data, which aligns with the main goal of this paper.

### Weaknesses

In contrast to the original augmentation graph discussed in HaoChen et al. (2021), the formulation discussed in this paper looks slightly strange and puzzling.

1. At the beginning of Section 3, $n$ is defined to be the number of training data, but soon after $n$ is assumed to be the domain cardinality $|\\mathcal{X}|$. Consequently, the training data covers the entire domain $\\mathcal{X}$ exactly, which is already a little bit unrealistic (but this may be fine if we focus only on the analysis of the expected loss).
2. In Section 3.3, $n_L + n_U = n$ is assumed, but this could also be strange because it is unnatural that the number of labeled and unlabeled data is exactly the same as the domain size $n = |\\mathcal{X}|$.
3. The authors observe the different behaviors of the eigenvalues and error bound depending on the relationship between $n_U$ and $k$, the representation dimension. I suppose that $n = n_L + n_U$ is the source of these behaviors, which would not be intrinsic to contrastive learning itself. Indeed, it could hardly happen that "when $k$ is larger than $n_U + r$, the lower bound of $\\lambda_{k+1}$ is unaffected by the noise rate" (Section 5.2). If it does happen, demonstrating that the downstream performance improves with larger $k$ empirically would be an important step for supporting the validity.
4. In Section 4.2, the posterior probability matrix for unlabeled data $\\mathbf{Y}_U$ is assumed to be a $(n_U \\times n_U)$-matrix, but I believe it should be a $(n_U \\times r)$-matrix. Accordingly, $\\tilde{\\mathbf{Y}}$ in equation (6) should be $(n_L + n_U) \\times r$. Moreover, $\\mathbf{Y}_U$ must not be an identity matrix because it is a posterior probability matrix and hence satisfies the sum-to-one constraint.

**Summary Of The Paper:**

In contrast to existing works on weakly supervised contrastive learning, this paper considers how weakly supervised information such as noisy labels and semi-supervised labels affects the downstream performance of contrastive learning. The theoretical analysis provided in this paper is based on the spectral contrastive loss and augmentation graph, a recent theoretical framework proposed by HaoChen et al. (2021). The authors incorporate weakly supervised information into the augmentation graph by using a convex combination of label and feature similarity adjacent matrices. As a result, it is shown that the downstream linear classification error can be improved unless the noise rate is moderate and the representation dimension is not very large.

**Summary Of The Review:**

The authors provide and demonstrate an interesting way to leverage weakly supervised information in contrastive learning by extending the framework of unsupervised contrastive learning. Nevertheless, there are several points that could be improved in the final version as pointed out in the above sections.

---

> ### Author Response · Authors · 2022-11-18
> **Response to Reviewer v3Am (2/2)**
>
> **Q5.** It's important to emphasize that this paper introduces an (apparently arbitrary) matrix to specify how to augment data. Another (slightly) confusing wording might be the section title of Section 4.1 "Similarity Graph Induced By Noisy Labels" (and Sections 4.2 and 4.3 as well). The graph does not seem to be induced from the underlying nature, but the authors specify a convenient one for the latter purpose as far as I understand.
>
> **A5.** *We clarify that our formulation of "similarity graph" is not a distributional assumption on the underlying similarity among data, but to formulate a possible probability of drawing positive samples in contrastive learning that takes both label and feature information into consideration. We address this point at the beginning of Section 4. Besides, following your suggestions, we change the word "induce" by "describing" throughout the paper.*
>
> ---
>
> **Q6.** In Section 5.3, the definition of $g_{f,B}(\bar{x})$ appearing in the fourth line looks a circular definition. The linear projection matrix $B$ should be involved in it.
>
> **A6.** *Thanks for pointing this out. Denote $B \in \mathbb{R}^{k\times r}$ as the weights of the downstream linear classifier. The linear predictor is denoted as $\bar{g}_{f, B}(\bar{x}) = \mathrm{arg max}_{i \in [r]} \mathrm{P}_{x\sim\mathcal{A}(\cdot|\bar{x})}(g_{f, B}(x)=i)$, where $g_{f,B}(x) = \mathrm{arg max}_{i \in [r]} (f(x)^\top B)_i$. We have fixed this typo in the revision.*
>
> ---
>
> **Q7.** In Section 5.3 (Assumption 2), many notations are not defined properly. It is essential to show what $y(x)$, $\hat{y}(x)$, and $\mathcal{P}_{\bar{X}}$ are. Moreover, $x_i$ and $y_i$ in equations (21) and (22) are not given yet.
>
> **A7.** *Here, $y(x)$ denotes the label of $x$ and $\hat{y}$ is a function for predicting labels of $x$, i.e. $\hat{y}: \mathcal{X} \to [r]$. $\mathcal{P}_{\bar{X}}$ is the probability distribution of original input data $\bar{x}$. We revise the typos $x_i$ and $y_i$ to $x$ and $y$, respectively. All of this has been fixed in the revision.*
>
> ---
>
> **Q8.** In Section 5.3 (Theorem 1), "$B^* \in \mathbb{R}^{r\times k}$ with norm $\|B^*\|_F \leq 1/\tilde{\lambda_k}$" is unclear. It seems that $\tilde{\lambda_k}$ has not been defined yet so far. In addition, it is ambiguous whether this norm bound is a newly-introduced assumption or a corollary of the assumptions so far.
>
> **A8.** *Thanks for pointing this out. Here is a typo that $\tilde{\lambda_k}$ should be $\lambda_k$, whose range can be derived from Proposition 2. It is a newly-introduced condition for Theorem 1 to hold.*
>
> ---
>
> **Q9.** Other typos.
>
> **A9.** *Thanks for your careful reading. We have fixed them in the revision.*
>
> ---
> Thanks for your constructive comments. Hope our explanations and additional experiments can address your concerns. Please let us know if there is more to clarify. We are happy to take your further questions during the rebuttal stage.

---

> ### Author Response · Authors · 2022-11-18
> **Response to Reviewer v3Am (1/2)**
>
> We express our sincere gratitude to Reviewer v3Am for appreciating the novelty of our work. We address your main concerns below.
>
> ---
> **Q1.** At the beginning of Section 3, $n$ is defined to be the number of training data, but soon after $n$ is assumed to be the domain cardinality $|\mathcal{X}|$. Consequently, the training data covers the entire domain $\mathcal{X}$ exactly, which is already a little bit unrealistic (but this may be fine if we focus only on the analysis of the expected loss).
>
> **A1.** *Thanks for pointing this out. Actually, there are two types of training samples in the paper, i.e. the **original input samples** and the **augmented samples**. Following your suggestion, we re-organize the notations as $N$ denoting the number of original input samples, and $n$ denoting the number of augmented samples. Following HaoChen et. al. (2021), $n$ is also the domain cardinality $|\mathcal{X}|$, which is exponentially large but finite. The reason is that we can generate different random augmentations to achieve sufficiently many augmented samples, which fill the entire domain.*
>
> ---
>
> **Q2.** In Section 3.3, $n_L+n_U=n$ is assumed, but this could also be strange because it is unnatural that the number of labeled and unlabeled data is exactly the same as the domain size $n=|\mathcal{\mathcal{X}}|$.
>
> **A2.** *Following HaoChen et. al. (2021), $n$ is both the domain cardinality $|\mathcal{X}|$ and the number of augmented samples, which is exponentially large but finite. The reason is that we can generate different random augmentations to achieve sufficiently many augmented samples, which fill the entire domain.*
>
> ---
>
> **Q3.** The authors observe the different behaviors of the eigenvalues and error bound depending on the relationship between $n_U$ and $k$, the representation dimension. I suppose that $n=n_L+n_U$ is the source of these behaviors, which would not be intrinsic to contrastive learning itself. Indeed, it could hardly happen that "when $k$ is larger than $n_U+r$, the lower bound of $\lambda_{k+1}$ is unaffected by the noise rate" (Section 5.2). If it does happen, demonstrating that the downstream performance improves with larger $k$ empirically would be an important step for supporting the validity.
>
> **A3.** *In fact, one cannot verify the magnitude of $\lambda_{k+1}$ through the downstream performance via changing $k$. Firstly, according to the value of $k$, $\lambda_{k+1}$ has different forms, whereas the magnitude of $\lambda_{k+1}$ relies on the eigenvalues of $\bar{\boldsymbol{A}}_0$ which is data specific but unknown.* *Secondly, the downstream error bound depends on both $\lambda_{k+1}$ (denominator) and noise rate (numerator), and therefore the trend of downstream performance cannot directly reflect the trend of $\lambda_{k+1}$ with respect to noise rate.*
>
> *Nonetheless, following your suggetions, we run additional experiments on CIFAR-10 with 30\% symmetric noise to show the influence of $k$ on downstream linear evaluation. We observe similar results regardless of output dimension $k$ in the following table.*
>
> | $k$ | 128 | 256 | 512 | 1024 |
> | :---- | :---: | :---: | :---: | :---: |
> |  Mix   | 91.89 | 92.18 | 92.12 | 91.77 |
>
> *Lastly, we are not sure if we get your concerns correctly. We are happy to discuss more if you could elaborate on this.*
>
> ---
>
> **Q4.** In Section 4.2, the posterior probability matrix for unlabeled data $\boldsymbol{Y}_U$ is assumed to be a $(n_U \times n_U)$-matrix, but I believe it should be a $(n_U \times r)$-matrix. Accordingly, $\tilde{\boldsymbol{Y}}$ in equation (6) should be $(n_L+n_U) \times r$. Moreover, $\boldsymbol{Y}_U$ must not be an identity matrix because it is a posterior probability matrix and hence satisfies the sum-to-one constraint.
>
> **A4.** *If you assume $\boldsymbol{Y}_U$ to be $n_U \times r$, it is equivalent to assume the correspondings labels are known as $r$ is the number of classes. Actually, in unsupervised contrastive learning, the representation of a sample is only made similar to that of its own augmentations, whereas made dissimilar to all the other augmented samples. This is equivalent to viewing the each unlabeled sample as having its own unique label. Therefore, $\boldsymbol{Y}_U$ is assumed to be an $n_U \times n_U$-identity matrix, which also satisfies the sum-to-one constaint. According to Equation (6), $\tilde{\boldsymbol{Y}}$ is naturally $(n_L+n_U) \times (r + n_U)$.*

---

### Official Review · Reviewer_kEnG · 2022-10-24

**Confidence:** 4
**Correctness:** 4
**Technical Novelty And Significance:** 2
**Empirical Novelty And Significance:** 2
**Recommendation:** 3

**Clarity, Quality, Novelty And Reproducibility:**

Novelty: The paper heavily follows HaoChen et al. (2021), which I feel is incremental work.

Reproducibility: no code or link is provided, so hardly to say.

**Strength And Weaknesses:**

This is a relatively interesting topic. The whole paper is well-written. The logic flow is very clear. Two citations should be considered in the Introduction together with Yan et al. (2022).
https://arxiv.org/pdf/2205.00186.pdf
https://arxiv.org/pdf/2108.04063.pdf

Overall, the experiments are weak. Only one dataset and two baselines are used. In table 1, the proposed method decreases as the others. So why not showing more results, e.g., step by 5% from 0 to 1? It will be more convincing to see the robustness of the proposed method. Why no result at 0%? SimCLR did you use InfoNCE? If so experiment part should include the eq. 32 by plugging in InfoNCE. Did you run 1000 epochs for all the experiments?


**Summary Of The Paper:**

This work aims at using weakly supervised information to improve contrastive learning. Following the work of HaoChen et al. (2021), this work also defines spectral contrastive loss and shows the graph under noisy and semi-supervised noisy conditions. After theoretical analysis, a mixed loss is verified on CIFAR-10.

**Summary Of The Review:**

The motivation is clear and the topic is interesting, but the work seems incremental. The experiment part is weak. It will be better to see more experiments.

---

> ### Author Response · Authors · 2022-11-18
> **Response to Reviewer kEnG**
>
> We thank Reviewer KEnG for acknowledging the topic and the clear presentation of our work. We address your main concerns below.
>
> ---
>
> **Q1.** Two citations should be considered in the Introduction together with Yan et al. (2022). https://arxiv.org/pdf/2205.00186.pdf https://arxiv.org/pdf/2108.04063.pdf
>
> **A1.** *We have added the citations in the Introduction "Recently, contrastive learning has been introduced to solve weakly supervised learning problems such as noisy label learning (Tan et al., 2021; Wang et al., 2022) and semi-supervised learning."*
>
> ---
>
> **Q2.** In table 1, why not showing more results, e.g., step by 5% from 0 to 1? It will be more convincing to see the robustness of the proposed method.
>
> **A2.** *Following your suggestions, we additional conducts experimental comparisons on CIFAR-10 with $\gamma$ varying from 0\% to 60\% in the following table. It shows that our Mix consistently outperforms SimCLR and SupCon.*
>
> | $\gamma$ | $0\%$ | $5\%$ | $10\%$ | $20\%$ | $30\%$ | $40\%$ | $50\%$ | $60\%$ |
> | :---- | :---: | :---: | :---: | :---: | :---: | :---: | :---: | :---: |
> | SimCLR | 92.30 | 91.99 | 91.63 | 91.59 | 90.95 | 90.42 | 90.18 | 88.92 |
> | SupCon | **95.48** | 93.86 | 92.62 | 90.56 | 87.76 | 84.80 | 80.56 | 74.90 |
> |  Mix   | **95.48** | **94.39** | **93.12** | **92.84** | **91.89** | **91.74** | **90.80** | **90.66** |
>
> ---
>
> **Q3.** Why no result at 0%?
>
> **A3.** *The performance for our method is exactly the same as SupCon when $\gamma=0\%$.*
>
> ---
>
> **Q4.** SimCLR did you use InfoNCE? If so experiment part should include the eq. 32 by plugging in InfoNCE.
>
> **A4.** *Yes, for SimCLR we use InfoNCE. Actually, we have already presented the results you required, because our Mix exactly use InfoNCE for both supervised and unsupervised loss in Eq. 32.*
>
> ---
>
> **Q5.** Did you run 1000 epochs for all the experiments?
>
> **A5.** *Yes, we run 1000 epochs for all experiments.*
>
> ---
>
> **Q6.** The paper heavily follows HaoChen et al. (2021), which I feel is incremental work.
>
> **A6.** *The main point of this paper is to investigate whether and when weakly supervised learning information helps improve contrastive learning. This paper is essentially different from HaoChen et. al. (2021) in the following aspects.*
> - [**Analyzing label information.**] *We **for the first time** establish a theoretical framework for weaky supervised contrastive learning, where we **translate the label information into a similarity graph**, whereas HaoChen et. al. (2021) analyzed the pure unsupervised contrastive learning.*
> - [**Eigenvalues of mixed graph.**] *The main technical difficulty of our analysis is to **discuss the eigenvalues of the mixed similarity graph containing both label and feature information (Proposition 2)**, rather than to utilize mathematical tools in HaoChen et. al. (2021).*
>
> *In summary, **the above two points are non-trivial and cannot be directly extended from HaoChen et. al. (2021).***
>
> ---
>
> Thanks for your constructive comments. Hope our explanations and additional experiments can address your concerns. Please let us know if there is more to clarify. We are happy to take your further questions during the rebuttal stage.

---

### Official Review · Reviewer_V7qm · 2022-10-25

**Confidence:** 4
**Correctness:** 4
**Technical Novelty And Significance:** 3
**Empirical Novelty And Significance:** 2
**Recommendation:** 6

**Clarity, Quality, Novelty And Reproducibility:**

### Clarity. Very good (mostly)
Other than the sloppy experiment setup, this paper is clearly written. Notations and concepts well defined and explained, nice discussion of background and related work.

### Quality. Mixed
Motivation, literature review and theoretical development are good, experiments need to be strengthened.

### Novelty. Okay
While the analyses is new, this work did not offer original recipes to guide the practice of weakly-supervised contrastive learning. Also the author(s) did not cover whether/how insights from this study can merit non-contrastive weakly-supervised learning.


**Strength And Weaknesses:**

### Strength
* This paper is well-written, relevant concepts and notations are clearly defined. Although much of content is very technically involved, the author(s) have accompanied these sections with remarks and analyses that are more accessible to non-expert audience.
* Studying the theoretical foundation for semi/weakly-supervised contrastive learning is a timely topic. This provides theoretical guidance most closely related to the popular pre-training + fine-tuning paradigm.
* Excellent discussion of the relevant literature, the coverage is adequate and up-to-date. That said, I think [A] will make a nice addition to the list, which discussed more generic settings of weak supervision.



### Weakness
* Experiments are a bit loose.
    * It is not clear how the author(s) make the labeled/unlabeled split in the experiments. I am assuming SimCLR and SupCon have used all data, and Mix also used all data both as labeled and unlabeled. If that is the case, then this is different from the setting used in the theoretical analysis, where labeled/unlabeled do not overlap.
    * Not an apple-to-apple comparison. The chosen baselines (SimCLR and SupCon) used the InfoNCE, while Mix used SpecNCE. It is well known that InfoNCE has some issues which motivates many follow up works [B, C, D, E], including SpecNCE. So the gain in Mix may not come from the theoretical predictions made in this paper, but rather gains from SepcNCE itself.
    * Only Cifar-10 results are reported, which is insufficient by current standards. The author(s) should at least include miniimagenet or tieredimagenet results.
    * The author(s) should also characterize the variability of the results, as the reported gains could be well within normal statistical fluctuations. While I understand pre-training can be costly, at least repeat the experiments on smaller datasets such as MNIST and report the standard deviation for the accuracy.
* To summarize the key take-away: (1) adding labeled data on top of self-supervised learning improves accuracy; (2) if the labels are too noisy, then it will wash away the information resulting no improvement. Although I appreciate the theoretical rigor, these are a bit straightforwardly intuitive and practitioners have long been practicing based on similar hunches.

#### References
[A] Strength from weakness: Fast learning using weak supervision. ICML, 2020

[B] Simpler, Faster, Stronger: Breaking The log-K Curse On Contrastive Learners With FlatNCE

[C] Provable guarantees for self-supervised deep learning with spectral contrastive loss. NeurIPS 2021

[D] Provable Stochastic Optimization for Global Contrastive Learning: Small Batch Does Not Harm Performance. ICML 2022

[E] EqCo: Equivalent Rules for Self-supervised Contrastive Learning


**Summary Of The Paper:**

++++++++ After Rebuttal +++++++++++++

In the rebuttal and follow up discussion, the author(s) have clarified my questions and provided additional empirical results to support their claims. I am adjusting the score accordingly.

(Author(s) please make sure the MIX loss are clearly defined in the revised manuscript.)

+++++++++++++++++++++++++++++++++

Building on the theoretical framework of the spectral contrastive learning, this paper studies the theory for weakly supervised contrastive learning. In particular, the author(s) presented a unified contrastive learning framework for two typical weakly supervised learning setups: noisy label learning and semi-supervised learning, and showed that under mild noise level both settings can improve the error bound of unsupervised contrastive learning with a carefully chosen hyper-parameter. Some empirical evidence is provided to support the claims.

**Summary Of The Review:**

Overall this paper is written with clarity. My major reservation is due to two points: (a) the key conclusions are aligned with intuition and they do not lend additional insights to better guide practice; (b) the comparison reported in the experiments is not aligned with the theory developed here (see my comments in the weakness, basically the author(s) used InfoNCE as baseline when they use have used SpecNCE).

I look forward to hearing the author(s) thoughts during the rebuttal discussion. Should more convincing argument or empirical results surface I will be happy to adjust my score.

---

> ### Author Response · Authors · 2022-11-18
> **Response to Reviewer V7qm**
>
> We express our sincere gratitude to Reviewer V7qm for appreciating the novelty and theoretical contribution of our work. We address your main concerns below.
>
> ---
>
> **Q1.** It is not clear how the author(s) make the labeled/unlabeled split in the experiments. I am assuming SimCLR and SupCon have used all data, and Mix also used all data both as labeled and unlabeled. If that is the case, then this is different from the setting used in the theoretical analysis, where labeled/unlabeled do not overlap.
>
> **A1.** *In our experiments, we do not make labeled/unlabeled split, i.e. all samples are with (noisy) labels. For SimCLR, we train on all samples ignoring their labels, whereas for Supcon and Mix, we train on all samples with their (noisy) labels.*
>
> ---
>
> **Q2.** Not an apple-to-apple comparison. The chosen baselines (SimCLR and SupCon) used the InfoNCE, while Mix used SpecNCE. It is well known that InfoNCE has some issues which motivates many follow up works [B, C, D, E], including SpecNCE. So the gain in Mix may not come from the theoretical predictions made in this paper, but rather gains from SepcNCE itself.
>
> **A2.** *In this paper, we use spectral contrastive loss (SpecNCE as you mentioned) for theoretical analysis, which inspires the Mix loss function in Eq. 32 to leverage both label and feature information. For fair comparisons in experiments, considering the standard SupCon method uses the InfoNCE loss, we also **use InfoNCE loss in experiments for standard SimCLR and our proposed Mix**. Therefore, **the gain in Mix comes exactly from the theoretically inspired loss function, rather than from the SpecNCE**.*
>
> ---
>
> **Q3.** Only Cifar-10 results are reported, which is insufficient by current standards. The author(s) should at least include miniimagenet or tieredimagenet results.
>
> **A3.** *Since miniimagenet and tieredimagenet datasets have different classes in their train and test sets, which is incompatible with out experiemntal setting, we run additional experiments on TinyImagenet-200, which contains 100000 images of 200 classes (500 for each class) downsized to 64×64 colored images. We show that Mix outperforms SimCLR and SupCon on TinyImagenet-200 in the following table.*
>
> | $\gamma$ | $0\%$ | $40\%$ |
> | :---- | :---: | :---: |
> | SimCLR | 53.78 $\pm$ 0.13 | 49.79 $\pm$ 0.05 |
> | SupCon | **63.79 $\pm$ 0.23** | 50.24 $\pm$ 0.09 |
> | Mix | **63.79 $\pm$ 0.23** | **53.40 $\pm$ 0.05** |
>
> ---
>
> **Q4.** The author(s) should also characterize the variability of the results, i.e. report the standard deviation for the accuracy.
>
> **A4.** *We run 3 trials for all experiments and present the mean and std in the following table. It shows that Mix gains significant performance improvement over SimCLR and SupCon.*
> | $\gamma$ | $0\%$ | $5\%$ | $20\%$ |
> | :---- | :---: | :---: | :---: |
> | SimCLR | 92.30 | 91.99 $\pm$ 0.06 | 91.59 $\pm$ 0.06 |
> | SupCon | **95.48** | 93.86 $\pm$ 0.05 | 90.56 $\pm$ 0.05 |
> |  Mix   | **95.48** | **94.39 $\pm$ 0.03** | **92.84 $\pm$ 0.09** |
>
> ---
>
> **Q5.** Although I appreciate the theoretical rigor, these are a bit straightforwardly intuitive and practitioners have long been practicing based on similar hunches.
>
> **A5.** *Despite the empirical experiences, **theoretical resutls reveal the learning mechanisms of contrastive learning with weakly supervised information, and provides the potential of using this knowledge for the design of new learning algorithms**. In this work, we for the first time establish a theoretical framework for weakly supervised learning contrastive learning. We prove that the downstream error of weakly supervised contrastive learning depends on the dimension $k$, label recovery rate $\delta$, and noise rate $\gamma$, given the eigenvalues of the unsupervised augmentation graph. **This reveals in what circumstances weakly supervised contrastive learning algorithm would work**. We believe our work could inspire more studies for deeper understandings of contrastive learning mechanism, and also possible new algorithms for weakly supervised learning tasks with contrastive learning approaches.*
>
> ---
>
> Thanks for your insightful and constructive comments. Hope our explanations and additional experiments can address your concerns. Please let us know if there is more to clarify. We are happy to take your further questions during the rebuttal stage.

---

> > ### Comment · Reviewer_V7qm · 2022-11-27
> > **Please clarify the Mix loss**
> >
> > Thanks for the rebuttal, I really appreciate the additional experiment results.
> >
> > However, I am now even more confused on how the Mix is actually implemented. Can you give more details on the statement "we also use InfoNCE loss in experiments for standard SimCLR and our proposed Mix"? Because I thought Mix is using the SpecNCE, isn't it? The $L_{sup}$ and $L_{unsup}$ in Eqn (32) are not clearly defined in the paper.

---

### Official Review · Reviewer_pQcs · 2023-01-07

**Confidence:** 5
**Clarity, Quality, Novelty And Reproducibility:** Overall, the paper is clearly written…
**Correctness:** 3
**Technical Novelty And Significance:** 3
**Empirical Novelty And Significance:** 2
**Recommendation:** 5

**Strength And Weaknesses:**

Strength:
--------------
Extending the theoretical framework on Haochen et.al. to weak supervision ~ by leveraging stochastic edges with prob \theta between samples.

using a combination of supcon and infoNCE is done before for semi-supervised setting (suNCEt:   https://arxiv.org/pdf/2006.10803.pdf )  and biased weak supervision (Positive Unlabeled Contrastive: https://arxiv.org/pdf/2206.01206.pdf )

Weakness :
----------------------
1. The empirical evaluation is limited - only on CIFAR-10 comparing infoNCE, supCon and the proposed approach.
I feel we need further empirical evidence over multiple datasets and under various noise levels.

2. This loss is very similar to "consistency regularization" where one would add a semantic consistency term (eg. infoNCE loss) with supervised loss (CE) see for example:
https://arxiv.org/pdf/2011.01403
https://arxiv.org/pdf/2106.08226
https://arxiv.org/pdf/2010.07835

Isn't this equivalent to https://arxiv.org/pdf/2011.01403 where instead of cvx comb of supcon and infoNCE they use cvx combination of CE (label consistency) and supcon / infoNCE etc (semantic consistency) -- since they tackle finetuning.

Isn't Mix just applying the same idea of enforcing semantic consistency by using label agnostic infoNCE ?

3. The parameter \theta is critical for the success of the method - it should be dependent on \gamma the noise rate. as pointed out by authors (and is the obvious intuition) that as noise rate is higher we should rely more on self supervision - which is label agnostic hence robust. what would be the dependence ? For example on Table 2 the authors show performance of Mix to be somewhat robust for different noise levels - it is critical for understanding the merit of the method to show the optimal \theta used for different values of \gamma. When does \theta_opt go to 0 - when all labels are corrupted ? a plot of \theta_optimal vs \gamma over multiple datasets is going to be helpful.
- 3.a. From Figure 1 it seems using Mix has little effect when \gamma is slightly high - for example at \gamma = 20% theta_opt = 0.1 whereas \gamma = 5% theta_opt = 0.9 so on slightly high gamma \theta -> 0 i.e. using infoNCE.

I am really ** Surprised ** by your Table 2 results especially the gap with infoNCE at large noise levels - which such small theta

4. Linear Probing: How did you linear probe ? on held out clean data ? or on noisy data. If on noise data - what loss did you use for linear probing ? since standard CE would result in performance degradation even if the encoder is high quality.
-- 4.a. As seen from Table 2 SimCLR results it seems you linear probed on noisy data since the infoNCE-encoder is identical for different \gamma.
-- 4.b How about you linear probe on clean data ? i.e. the encoder is trained on noisy supervision but the linear head is trained with clean



**Summary Of The Paper:**

Contrastive Learning (CL) has become popular for both unsupervised and fully supervised settings. Recent empirical studies have indicated that CL can also be beneficial under weakly (ex noisy label) or semi-supervised settings.

This paper studies CL under weakly supervised setting and proposed an approach Mix where they propose to use a convex combination of unsupervised (infoNCE) and supervised contrastive loss (supcon): \theta infoNCE + (1 - \theta) supcon.

They analyze this algorithm using tools from spectral contrastive learning Haochen et.al. and show some empirical validation comparing with supcon and infoNCE on cifar and imagenet benchmark.

**Summary Of The Review:**

The theoretical contributions are somewhat novel , but I feel the paper lacks algorithmic novelty (or rather they do not position the contribution w.r.t relevant literature ex - semi supervised , consistency regularization literature where similar algorithm has been used).
Empirical evidence is somewhat lacking - especially does not flesh out the strategy to select theta , make this algorithm work well in reality.

Firtsly, cheers to the authors on tackling this important problem in a systematic way -  Overall, I feel the authors might want to dive deeper and investigate further and submit a stronger paper in next ML venue - will be looking forward to the findings.

---

### Author Response · Authors · 2022-11-18
**A Summary of Paper Updates**

We thank all reviewers for their constructive comments. We have updated the paper accordingly with the following major changes:

- Section 6, Table 1. Add standard deviations for results with noise rate 5\% and 20\%.

- Appendix A.3, Table 2. Add more experimental results with noise rate $\gamma$ varying from 0\% to 60\%.

- Appendix A.3, Table 3. Add experimental comparisons on the TinyImagenet-200 dataset.

- Add more explanations to make the mathematical expressions clear.

---

### Decision · Program_Chairs · 2023-01-20

**Decision:**

Reject

**Justification For Why Not Higher Score:**

The conclusion was that neither the theory nor the experiments are strong enough to stand on their own for acceptance. Namely, while there are some new elements, a lot of the theory borrows from the spectral contrastive loss paper by HaoChen et al. (and several reviewers pointed out some rather strong and unrealistic additional assumptions for the setup considered); the experiments were fairly small scale and somewhat uncarefully carried out and written.

**Justification For Why Not Lower Score:**

N/A

**Metareview: Summary, Strengths And Weaknesses:**

The paper concerns an analysis of how weak supervision (e.g. noisy labels) can be integrated with contrastive techniques (specifically, spectral properties of the augmentation graph), to provide classification error bounds. There was some back and forth between the reviewers and the authors, but ultimately the conclusion was that neither the theory nor the experiments are strong enough to stand on their own for acceptance. Namely, while there are some new elements, a lot of the theory borrows from the spectral contrastive loss paper by HaoChen et al. (and several reviewers pointed out some rather strong and unrealistic additional assumptions for the setup considered); the experiments were fairly small scale and somewhat uncarefully carried out and written.